# Selective Degradation Permits a Feedback Loop Controlling Annexin A6 and Cholesterol Levels in Endolysosomes of NPC1 Mutant Cells

**DOI:** 10.3390/cells9051152

**Published:** 2020-05-07

**Authors:** Elsa Meneses-Salas, Ana García-Melero, Patricia Blanco-Muñoz, Jaimy Jose, Marie-Sophie Brenner, Albert Lu, Francesc Tebar, Thomas Grewal, Carles Rentero, Carlos Enrich

**Affiliations:** 1Departament de Biomedicina, Unitat de Biologia Cel·lular, Facultat de Medicina i Ciències de la Salut, Universitat de Barcelona, 08036 Barcelona, Spain; elsameneses.s@gmail.com (E.M.-S.); anagarciamelero@gmail.com (A.G.-M.); patrii__4@live.com (P.B.-M.); tebar@ub.edu (F.T.); 2Centre de Recerca Biomèdica CELLEX, Institut d’Investigacions Biomèdiques August Pi i Sunyer (IDIBAPS), 08036-Barcelona, Spain; 3School of Pharmacy, Faculty of Medicine and Health, University of Sydney, Sydney 2006, NSW, Australia; jjos6151@uni.sydney.edu.au (J.J.); marie-sophie.brenner@sydney.edu.au (M.-S.B.); 4Department of Biochemistry, Stanford University School of Medicine, Stanford, CA 94305, USA; alulopez@stanford.edu

**Keywords:** cholesterol, AnxA6, chaperone-mediated autophagy, endolysosomes, NPC1, Lamp2A

## Abstract

We recently identified elevated annexin A6 (AnxA6) protein levels in Niemann–Pick-type C1 (NPC1) mutant cells. In these cells, AnxA6 depletion rescued the cholesterol accumulation associated with NPC1 deficiency. Here, we demonstrate that elevated AnxA6 protein levels in NPC1 mutants or upon pharmacological NPC1 inhibition, using U18666A, were not due to upregulated AnxA6 mRNA expression, but caused by defects in AnxA6 protein degradation. Two KFERQ-motifs are believed to target AnxA6 to lysosomes for chaperone-mediated autophagy (CMA), and we hypothesized that the cholesterol accumulation in endolysosomes (LE/Lys) triggered by the NPC1 inhibition could interfere with the CMA pathway. Therefore, AnxA6 protein amounts and cholesterol levels in the LE/Lys (LE-Chol) compartment were analyzed in NPC1 mutant cells ectopically expressing lysosome-associated membrane protein 2A (Lamp2A), which is well known to induce the CMA pathway. Strikingly, AnxA6 protein amounts were strongly decreased and coincided with significantly reduced LE-Chol levels in NPC1 mutant cells upon Lamp2A overexpression. Therefore, these findings suggest Lamp2A-mediated restoration of CMA in NPC1 mutant cells to lower LE-Chol levels with concomitant lysosomal AnxA6 degradation. Collectively, we propose CMA to permit a feedback loop between AnxA6 and cholesterol levels in LE/Lys, encompassing a novel mechanism for regulating cholesterol homeostasis in NPC1 disease.

## 1. Introduction

Cholesterol is a fundamental lipid in mammalian cells responsible for the proper organization and function of membranes. The levels of cholesterol in cellular compartments are tightly controlled through *de novo* synthesis in the endoplasmic reticulum (ER), and the uptake of low-density lipoproteins (LDL) by receptor-mediated endocytosis. As excess amounts of cellular unesterified (free) cholesterol are cytotoxic, cells have developed sophisticated circuits to regulate its intracellular sorting, trafficking and storage [1]. Once internalized, LDL-derived cholesterol is targeted to the LE/Lys compartment where cholesterol is first transferred from intraluminal vesicles (ILVs) to the limiting membrane via NPC2, lysobisphosphatidic acid (LBPA), and possibly other transporters [2,3,4,5]. In the outer LE/Lys membrane, NPC1 is the major transporter, and together with several other cholesterol-binding proteins [6], is responsible for LE-Chol export and subsequent transfer to other cellular destinations [7], preferentially the plasma membrane and ER, but also mitochondria, peroxisomes, Golgi, or recycling endosomes. In the ER, cholesterol can be re-esterified, permitting cytoplasmic storage of excess cholesterol in lipid droplets. Several pathways regulate the delivery of cholesterol from LE/Lys to other cellular sites. This includes vesicular trafficking via small GTPases (e.g., Rab7, Rab8, and Rab9), non-vesicular transport mediated by lipid transfer proteins, or cholesterol transfer across membrane contact sites (MCS) [8]. In addition, autophagy also contributes to regulate lipid metabolism in the LE/Lys compartment [9,10,11]. Therefore, it has been suggested that alterations in autophagy may contribute to the pathology of lipid storage disorders. For example, Sarkar et al. (2013) identified defective autophagy in Niemann–Pick type C1 (NPC1) disease models to be associated with cholesterol accumulation [12]. In these studies, failure of the SNAP receptor (SNARE) machinery caused defects in amphisome formation, which impaired the maturation of autophagosomes, while the lysosomal proteolytic function remained unaffected. In this setting, ectopic NPC1 expression rescued the defect in autophagosome formation. Intriguingly, both the inhibition and stimulation of autophagy caused cholesterol accumulation in LE/Lys, suggesting that the regulation of autophagy may be intimately linked to changes in LE-Chol levels [13,14]. To date, the precise way in which autophagy can alter LE-Chol homeostasis still remains elusive.

The complexity of autophagic pathways has been described in detail in recent reviews [15,16]. Calcium (Ca^2+^) is a well-known regulator of autophagy, yet despite the wide range of lysosomal storage diseases that share defects in both autophagy and Ca^2+^ homeostasis, the intersection between these two pathways is still not well characterized [17]. In fact, a number of Ca^2+^-binding proteins, including apoptosis-linked gene-2 (ALG-2); calmodulin; several S100 family proteins; ALG-2-interacting protein 1 (AIP1, also called Alix); calcineurin; as well as Ca^2+^ channels in LE/Lys, the ER, or mitochondria [18], have been associated with autophagy.

In addition, three members of the annexin family—AnxA1, A2, and A5—have been associated with autophagic processes [19]. Annexins are a conserved multigene family of proteins that bind to membranes in a Ca^2+^-dependent manner and are widely expressed [20]. Within the endocytic pathway, they have been associated with a variety of membrane trafficking events, including vesicle transport and fusion, microdomain organization, and LE/Lys positioning, as well as membrane-associated actin cytoskeleton dynamics and cholesterol homeostasis [21,22,23]. Furthermore, AnxA1 and AnxA6 participate in MCS formation [24,25], regulating the transfer of cholesterol, and possibly other lipids and Ca^2+^, from LE/Lys to other cellular sites [23].

Despite the accumulating knowledge on the abovementioned annexins and their mode of action in late endocytic circuits, including autophagy, our understanding how these annexins operate in this cellular location is still incomplete. Yet, to exert their various functions, their physical association with the LE/Lys compartment seems essential. The accessibility to membrane lipids that serve as annexin binding sites, in particular, phosphatidylserine and phosphatidic acid, but also cholesterol and phosphatidylinositol (4,5)-bisphosphate (PIP2), is well documented [22]. Therefore, for the association of annexins with LE/Lys membranes in space and time, the lipid composition of LE/Lys, which is likely to undergo dynamic changes not only due to membrane turnover, but also nutrient availability and consequently, the endocytic activity of cells, together with the differential affinity of each annexin for individual lipid species, is most likely critical. In addition, the cytosolic tails of proteins in the limiting LE/Lys membrane may directly or indirectly further add to establish interactions. In support of this, direct protein–protein interactions of annexins with several LE/Lys proteins have been reported, including the two-pore channel (TPC1/2) [26], NPC1 [27], or the vacuolar proton pump (ATPase H^+^ transporting VO subunit a2) [28]. This complex interactome of annexins with lipids and proteins could ensure proper LE/Lys functioning and provide opportunity to control the intracellular movement of lipids, in particular, cholesterol, into and out of the LE/Lys compartment. Indeed, we and others demonstrated that AnxA1, A2, A6, and A8 contribute to cholesterol trafficking along endo- and exocytic pathways, suggesting that this subset of annexins, as well as other Ca^2+^-binding proteins, are good candidates for achieving regulatory tasks not only in regard to cellular cholesterol homeostasis, but also cholesterol-dependent activities in LE/Lys such as autophagy [23].

A recent proteomic analysis of CMA-targeting motifs supported earlier studies that revealed most members of the human annexin family to contain the KFERQ-like motif known to target cytosolic proteins for CMA or endosomal microautophagy (eMi) into lysosomes [15,29,30,31]. CMA and eMi are initiated by the binding of heat shock cognate 71 kDa protein (hsc70) to the pentapeptide motif in cytosolic proteins, which enables their binding to Lamp2A at the LE/Lys membrane. This initial recruitment step is critical for the subsequent translocation of these selective substrates into lysosomes via CMA. eMi is less well studied and relies on the endosomal sorting complexes required for transport (ESCRT) machinery [32]. Besides, CMA and eMI, macroautophagy can also engulf cytoplasmic proteins and organelles for lysosomal degradation. This process requires the formation of autophagosomes before fusion with lysosomes. Several annexins, including AnxA1 and AnxA5, have been suggested to play a role in autophagosome maturation or further steps of autophagy [33], but the underlying molecular mechanisms still have to be elucidated.

We previously demonstrated that overexpression of AnxA6 was associated with LE-Chol accumulation, mimicking a NPC1 mutant-like phenotype, which was detrimental for the intracellular trafficking along endo- and exocytic routes [27,34]. In addition, we recently documented that NPC1 mutant Chinese hamster ovary (CHO) cells displayed elevated AnxA6 levels. This coincided with an increased and AnxA6-mediated recruitment of a Rab7-GTPase activating protein (Rab7-GAP), TBC1D15, to LE/Lys to inactivate Rab7. AnxA6-dependent Rab7 inactivation in NPC1 mutant cells was associated with a reduction of MCS between LE/Lys and the ER. *Vice versa*, AnxA6 depletion in NPC1 mutant cells caused elevated Rab7 activity, increased MCS formation, and significantly reduced the amount of cholesterol in LE/Lys [25].

In this report, we examined the link between AnxA6 protein levels and the extent of LE-Chol accumulation in LE/Lys of NPC1 mutant cells. Our findings reveal a complex feedback loop involving CMA that controls AnxA6 and LE-Chol levels. In NPC1 deficiency, or other disease settings where lysosomal dysfunction may be associated with LE-Chol accumulation, this may eventually increase the amount of AnxA6 in ILVs of the LE/Lys compartment. Remarkably, these findings could be relevant for intercellular communication, as extracellular vesicles, or exosomes, which deliver biologically active molecules to recipient cells, originate from the LE/Lys compartment, and could explain elevated levels of this annexin in exosomes observed in other pathologies [35,36].

## 2. Material and Methods

### 2.1. Cells, Reagents, Cell Culture and Transfections

CHO cells were grown in Ham’s F-12 with 10% fetal calf serum (FCS), L-glutamine (2 mM), penicillin (100 units/mL), and streptomycin (100 μg/mL) at 37 °C, 5% CO_2_. CHO M12 and CHO 2-2 were kindly provided by Drs. L. Liscum (Tufts University School of Medicine, Boston, MA, USA) and D. Ory (Washington University, St. Louis, MO, USA). The AnxA6-depleted CHO M12-A6ko cell line was generated in our laboratories using CRISPR/Cas9 editing technology as described [25]. Control (GM5659D) and NPC1 mutant human skin fibroblasts (GM03123) were from the Coriell Institute for Medical Research (Camden, NJ, USA) and cultured in DMEM, 10% FCS, L-glutamine (2 mM), penicillin (100 units/mL), and streptomycin (100 μg/mL) at 37 °C, 5% CO_2_. Nutrient Mixture Ham’s F-12 and DMEM were from Biological Industries (Cromwell, CT, USA). Hank’s buffered salt solution (HBSS) was from Gibco (Waltham, MA, USA). Filipin, saponin, U18666A, cycloheximide (CHX), leupeptin, and aprotinin were from Sigma-Aldrich (St. Louis, MO, USA). Paraformaldehyde (PFA) was from Electron Microscopy Sciences (Hatfield, PA, USA), and Mowiol was from Calbiochem (San Diego, CA, USA). Glutathione S-transferase (GST) (GE Healthcare, Chicago, IL, USA) and GST-fusion protein (Rab Interacting Lysosomal Protein (RILP)-C33-GST) were produced in *E. coli* BL21 cells and purified using Glutathione-Sepharose 4B beads (GE Healthcare, Chicago, IL, USA) as reported previously [37]. For transient transfections, cells were transfected using GenJet (SigmaGen Laboratories, Rockville, MD, USA) following manufacturer’s instructions.

### 2.2. RNA Extraction and Quantitative Real-Time PCR

Total RNA was extracted using the RNeasy Mini Kit (Qiagen, Venlo, Netherlands) in accordance with the manufacturer’s protocol. One microgram of RNA was reverse transcribed using High-Capacity cDNA Reverse Transcription Kit (Applied Bioscience, Waltham, MA, USA). In a final volume of 20 μL, real-time PCR Brilliant SYBRGreen QPCR Master Mix (Agilent Technologies, Santa Clara, CA, USA) and 10 μL of 1:20 diluted cDNA was used as a template for PCR analysis using the LightCycler system (Roche Diagnostics, Rotkreuz, Switzerland). Specific hamster primers (*AnxA6*: forward 5′-ccgggaagatgctaggaat-3′, reverse 5′-accctggtgagggtcttctt-3′; *Rpl13*: forward 5′-gccccacttccacaaggatt-3′, reverse 5′-ataccagccaccctgagttc-3′) were added and a standard PCR amplification protocol (10 min at 95 °C, 45 cycles of 30 s. at 95 °C, 15 s. at 60 °C and 30 s. at 72 °C, 10 s. at 95 °C and 60 s. at 65 °C) was performed according to manufacturer’s instructions. Values were normalized to the *Rpl13* gene in each sample.

### 2.3. Immunoblotting

For western blotting, whole cell lysates were prepared in lysis buffer (50 mM Tris, 150 mM NaCl, 5 mM EDTA, pH 7.5), protease inhibitors (1 mM Na_3_VO_4_, 10 mM NaF, 1 mM phenylmethylsulfonyl fluoride, 10 µg/mL leupeptin, and 10 µg/mL aprotinin), and equal amounts of protein, as determined by Bradford assay, were separated by 10% SDS-PAGE and transferred to a Hybridation Nitrocellulose membrane (Millipore, Burlington, MA, USA). After blocking in 5% skimmed milk, western blots were performed using rabbit anti-AnxA6 serum [38], mouse anti-β-actin (MP Biomedicals, Illkirch-Graffenstaden, France), mouse anti-epidermal growth factor receptor kinase substrate 8 (Eps8; BD Transduction Lab, San Jose, CA, USA), rabbit anti-GFP (Abcam, Cambridge, UK), rabbit anti-Rab7 (Cell Signaling Technology, Danvers, MA, USA) and mouse anti-α-tubulin (Sigma-Aldrich, St. Louis, MO, USA). Appropriate peroxidase-conjugated secondary antibodies (BioRad Laboratories, Hercules, CA, USA) and enhanced detection (EZ-ECL; Enhanced ChemiLuminiscence, Biological Industries, Cromwell, CT, USA) for band visualization were used. The intensity of bands was quantified using ImageJ and results were normalized to actin.

### 2.4. Rab7-GTP Pull-Down Assays

Cells were solubilized in pull-down buffer (50 mM Tris, 150 mM NaCl, 1% Triton X-100, 0.1 mM CaCl_2_, pH 7.3) supplemented with protease/phosphatase inhibitor cocktail (see above). Samples were centrifuged at 12,000× *g* for 10 min at 4 °C. Proteins from post-nuclear supernatants (400–700 µg) were incubated with Glutathione Sepharose 4B beads coated with purified recombinant RILP-C33-GST (40–70 µg) fusion protein for 2 h at 4 °C. GST was used as a negative control (data not shown). Samples were washed 3 times, collected in 30 µL of 1× loading buffer, and analyzed by western blotting (see above).

### 2.5. Fluorescence Microscopy

For fluorescence studies, CHO M12 cells were grown on coverslips, transfected with Lamp2A-GFP as per manufacturer instructions, and after 48 h, fixed with 4% PFA for 20 min, washed and then stained with filipin (0.05 mg/mL) and mounted in Mowiol as described [39]. Samples were observed using a Leica TCS SP5 laser scanning confocal microscope equipped with a DMI6000 inverted microscope, blue diode (405 nm), Argon (458/476/488/496/514), diode pumped solid state (561 nm), HeNe (594/633 nm) lasers, and APO 63× oil immersion objective lenses (Leica Microsystems, Wetzlar, Germany). Image analysis was performed with NIH ImageJ software as described below [40].

The quantification of filipin staining per cell (≥20 cells per group) was performed as described [25] and served to determine the relative amounts of free cholesterol in enlarged, perinuclear cholesterol-enriched and Lamp2A-positive late endosomal vesicles in NPC1 mutant M12 cells.

### 2.6. Transmission Electron Microscopy (TEM)

CHO-WT and CHO M12 cells were fixed at 4 °C with a mixture of 4% PFA and 0.1% glutaraldehyde in 0.1 M phosphate buffer, then scrapped, pelleted, washed in 2% PFA, then cryofixed by high pressure freezing (EM HPM100; Leica Microsystems, Wetzlar, Germany) and finally freeze-substituted for 80 h at −90 °C in anhydrous acetone containing 0.5% uranyl acetate. Next, samples were embedded in Lowicryl HM20 resin (Electron Microscopy Science, Hatfield, PA, USA). Ultrathin sections were obtained using an Ultracut UC6 ultramicrotome (Leica Microsystems, Wetzlar, Germany) and picked up on Formvar-coated nickel grids and observed using a transmission electron microscope, JEOL JEM-1010 (JEOL USA, Peabody, MA, USA) fitted with a Gatan Orius SC1000 (model 832) digital camera (Gatan Inc, Pleasnaton, CA, USA).

For immunolabeling, sections were incubated at room temperature on drops of 5% bovine serum albumin (BSA) in PBS for 20 min, followed by anti-AnxA6 polyclonal antibody (1:5 dilution) in PBS for 1 h. Then, sections were incubated for 30 min using anti-rabbit 12 nm colloidal gold (Jackson ImmunoResearch, West Grove, PA, USA) using a 1:30 dilution in PBS/1% BSA. This was followed by five washes with drops of PBS and 5 min in 1% glutaraldehyde; then, grids were rinsed with milliQ water and air dried. As a control for non-specific binding of the colloidal gold-conjugated antibody, the primary AnxA6 antibody was omitted. Three independent experiments were analyzed and at least 2 grids were used for each condition per cell line. The minimum number of cells scored for each condition was 25 and the average number of sections (fields) was 40. Data are shown as the mean ± SEM.

### 2.7. Image Analysis

Image analysis was performed using NIH ImageJ software [40]. When comparing different samples, images were captured and systematically screened using identical microscope settings.

### 2.8. Bulk Degradation Experiments

For the assessment of bulk degradation, CHO-WT, CHO M12, and CHO M12-A6ko cells were incubated with 10 µg/mL Dye Quenched-Red-BSA (DQ-Red-BSA) (Thermo Fisher Scientific, Waltham, MA, USA) in complete culture medium for 6 h at 37 °C. Then, cells were fixed in 4% PFA, washed extensively, stained with 4′,6-diamidino-2-phenylindole (DAPI) (Sigma-Aldrich, St. Louis, MO, USA) for 20 min, and then mounted with Mowiol. Control experiments with 100 μM Bafilomycin A1 (Sigma-Aldrich, St. Louis, MO, USA), which blocks lysosomal acidification, were performed (data not shown) [41,42]. Images were obtained with a Zeiss LSM880 Confocal Microscope (Zeiss Microscopy, Jena, Germany). Quantification of fluorescence intensity was performed as described [41].

### 2.9. Statistics

Unless mentioned in the figure legend, group data are presented as mean ± SEM. Comparison between 2 groups was analyzed by Student’s *t*-test, comparison between more than 2 groups was analyzed by one-way ANOVA with a Bonferroni *post hoc* test, and comparison between groups was analyzed using Prism software (GraphPad Software, San Diego, CA, USA). Differences were considered statistically significant at * *p* < 0.05, ** *p* < 0.01, *** *p* < 0.001.

## 3. Results and Discussion

### 3.1. AnxA6 Expression in NPC1 Mutant Cells

Cholesterol accumulation in LE/Lys is a distinctive feature of cells carrying loss-of-function mutations in the NPC1 gene and we previously showed that AnxA6 overexpression mimics this phenotype [27]. Furthermore, we recently identified that endogenous AnxA6 levels were significantly elevated in CHO cell lines harboring different NPC1 mutations (M12, 2-2) [25]. This prompted us to investigate the relationship between AnxA6 levels and LE-Chol accumulation in the LE/Lys compartment of NPC1 mutant cells. We first aimed to validate these findings by analyzing AnxA6 protein levels in human control (HSF-WT) and NPC1 patient skin fibroblasts (GM03123), or upon incubation of CHO wild type (WT) cells with the pharmacological NPC1 inhibitor U18666A. Indeed, loss of NPC1 function in both settings correlated with increased amounts of AnxA6 (Figure 1A,B). Remarkably, higher AnxA6 protein levels in cells lacking functional NPC1 protein were not associated with a significant increase in AnxA6 mRNA expression, as shown by quantitative RT-PCR of NPC1 mutant CHO cells or after U18666A treatment (Figure 1C). Therefore, we reasoned that alterations in the posttranscriptional regulation could be responsible for elevated AnxA6 protein levels in NPC1 mutant cells, including defects in AnxA6 degradation and/or AnxA6 protein accumulation in the LE/Lys compartment due to lysosomal dysfunction in NPC disease [43].

Therefore, we examined the bulk degradation using fluorescently-labeled DQ-Red-BSA as cargo [41], which is internalized and transported to lysosomes and then cleaved, resulting in a bright red fluorescent signal. In line with published data [12], CHO M12 cells displayed a 2-3-fold increased capacity of degradative endolysosomes (Appendix A, see quantification in B) compared to CHO-WT cells. Strikingly, AnxA6 depletion in CHO M12 cells (CHO M12-A6ko), which we reported recently to significantly reduce LE-Chol accumulation in these cells [25], also drastically reduced lysosomal DQ-Red-BSA degradation comparable to the CHO-WT control cells. It is tempting to speculate that in addition to the Rab7-mediated restoration of cholesterol export in NPC1 mutant cells upon AnxA6 depletion [25], the reduced bulk degradation in CHO M12-A6ko cells is also connected to elevated Rab7 activity in these cells, as Rab7 is well known to drive the fusion of late endosomes with lysosomes and cargo degradation [44]. Furthermore, as this connects AnxA6 up- or downregulation to the modulation of endolysosomal degradation in NPC1 mutant cells, one could envisage also other degradation routes such as autophagy, to be altered by changes in AnxA6 levels. Yet, as described further below, degradation can be selective and different pathways and mechanisms have been described.

In some models, lipid accumulation in LE/Lys inhibits lysosomal cathepsins, leading to impaired degradation of lysosomal cargo and increased autophagic intermediates [14]. However, in the latter experiments described above (Appendix A), DQ-Red-BSA degradation was increased in CHO M12 cells, and we hypothesize that possibly the enlarged LE/Lys compartment in these NPC1 mutant cells [45,46] may be one of the underlying causes for this observation. We have previously reported a reduction in the size of the LE/Lys compartment in AnxA6-depleted NPC1 mutant cells [25], which might reduce their capacity to process DQ-Red-BSA to levels comparable to those observed in CHO-WT cells. Yet, diverse outcomes when studying lysosomal functions in NPC1 mutant cells or other lysosomal storage diseases [47] have been reported. Therefore, additional factors to consider include the differential and cell-specific repertoire of other players driving LE/Lys function, but also the possibility that degradation is selective [12], which is possibly relevant here, could be sensitive to the amount of cholesterol or other lipids in the LE/Lys compartment [48]. Along these lines, several studies have implicated increased LE-Chol as a decisive factor for the regulation of LE functioning such as ILV formation and/or the sorting of LE/multivesicular bodies (MVB)/Lys structures for autophagic pathways such as CMA and eMi or exocytosis [15,49,50]. Moreover, autophagy delivers cholesterol to the lysosome, creating a feedback loop that promotes further lipid storage and lysosomal dysfunction [14,51]. Thus, although accumulating evidence points at LE-Chol accumulation interfering with the proper functioning of a variety of autophagic and lysosomal tasks, more research is needed to identify the factors or environmental changes that create sensitivity towards LE-Chol and/or AnxA6 levels for the various lysosomal activities.

### 3.2. Selective Degradation of AnxA6 in LE/Lys

As previous reports were in support of AnxA6 being targeted for lysosomal degradation by selective autophagy (CMA/eMi) [30,50], we next examined if LE-Chol accumulation in M12 cells could compromise AnxA6 degradation. For this purpose, CHO-WT and CHO M12 cells were incubated with CHX alone to inhibit protein synthesis, or in combination with leupeptin, which blocks lysosomal degradation. Given the slow protein turnover of AnxA6 [30], cells were incubated with CHX ± leupeptin for 48 h. Then, whole cell lysates were prepared and protein levels of AnxA6 and Eps8, which also contains a pentapeptide-motif and is degraded via CMA, were determined by Western blotting (Figure 2A see quantification in Figure 2B,C). CHX-treated CHO-WT cells displayed only a minor reduction in AnxA6 and Eps8 protein levels, probably reflecting the long half-life of these proteins (>48 h) [30] (compare lanes 1 and 2). Yet, the combined treatment of CHX and leupeptin in CHO-WT cells resulted in AnxA6, as well as Eps8, protein levels that were higher compared to their respective negative controls (compare lane 1 and 3, for quantification see Figure 2B,C), confirming that this assay was suitable to examine lysosomal protein degradation. However, the combined treatment of CHX and leupeptin resulted in AnxA6 and Eps8 protein levels similar to CHO M12 controls (Figure 2A, compare lanes 4 and 6). Likewise and consistent with the results described previously [25] and above (see Figure 1), AnxA6 protein levels in CHO M12 cells were substantially elevated compared to CHO-WT cells (compare lanes 1 and 4; for quantification see Figure 2B,C). Thus, the overall behavior of the CMA marker protein, Eps8, as well as AnxA6 protein pools, in the presence of inhibitors blocking protein synthesis and lysosomal degradation, showed similar trends in both cell lines. Yet, when comparing the response of CHO-WT and CHO M12 cells to leupeptin, CHO-WT showed a much stronger elevation of Eps8 and AnxA6 protein levels compared to CHO M12 cells (compare lanes 1 and 3 with 4 and 6), indicating that in NPC1 mutant cells the degradation route of Eps8, but also AnxA6, most likely via CMA, was inhibited.

To further assess if an inhibition of CMA-mediated AnxA6 degradation could explain upregulated AnxA6 levels in NPC1 mutant cells, we next aimed to activate this pathway through the overexpression of Lamp2A, which acts as a receptor for CMA substrates and thereby activates the CMA pathway [49,50]. Therefore, GFP-tagged Lamp2A was ectopically expressed in CHO-WT and CHO M12 cells and cell lysates were then analyzed for AnxA6 expression levels by Western blotting. As shown in Figure 3A (see quantification in 3B), overexpression of Lamp2A-GFP caused a significant decrease of AnxA6 levels in NPC1 mutant cells, further supporting AnxA6-mediated degradation via the CMA pathway and its reduced efficacy due to LE-Chol enrichment. Moreover, pointing at a link between levels of AnxA6 and LE-Chol, Lamp2A-induced downregulation of AnxA6 was associated with increased Rab7 activity, as judged by elevated Rab7-GTP levels in CHO M12 and CHO 2-2 cells in pulldown assays as described (Figure 3C). This is in agreement with our recent findings that identified AnxA6 depletion to prevent the recruitment of the Rab7-GAP TBC1D15 to LE/Lys [25]. In NPC1 mutant cells, this led to Rab7 activation and was accompanied by the restoration of LE-Chol export in these cells. The latter observation further suggests that the reduced LE-Chol accumulation in cells lacking NPC1 normalizes CMA-mediated AnxA6 degradation.

### 3.3. Lamp2A Regulates Cholesterol Levels in Endolysosomes of NPC1 Mutant Cells

Lamp2A is a lysosomal-associated membrane protein that acts as a protecting barrier for hydrolases and a receptor for CMA substrates, but has also been implicated in cholesterol export from LE/Lys [52,53,54]. Inversely, cholesterol levels may influence Lamp2A protein conformation or interaction with other lysosomal partners, with consequences for autophagy and cellular metabolism [52]. As mentioned above, ectopic Lamp2A expression in NPC1 mutant cells led to AnxA6 downregulation and elevation of Rab7-GTP levels (Figure 1 and Figure 2), a scenario that we showed was associated with restoration of LE-Chol export in cells lacking NPC1 [25]. This prompted us to study whether Lamp2A-induced CMA activation possibly correlated with diminished LE-Chol accumulation in NPC1 mutant cells. Therefore, Lamp2A-GFP was ectopically expressed in NPC1 mutant CHO M12 cells and to examine LE-Chol accumulation, cells were then fixed and stained with filipin to visualize unesterified (free) cholesterol (Figure 4). Remarkably, quantification of filipin staining per cell, which resembles almost exclusively prominent and enlarged, perinuclear cholesterol-enriched, and Lamp2A-positive late endosomal vesicles in NPC1 mutant M12 cells (see also enlarged inserts in 4C), identified a significant decrease of filipin intensity per cell by 15% in CHO M12 cells (9.474 ± 0.3 vs. 8.059 ± 0.45) ectopically expressing Lamp2A compared to the neighboring non-transfected CHO M12 cells (Figure 4A, see quantification in Figure 4B). Indeed, enlarged images revealed a mixed population of Lamp2A-positive vesicles, many of them only weakly stained with filipin, indicating restoration of LE-Chol export (Figure 4C; see white arrows in the enlarged inset and line profile intensity plot). Taken together, it appears feasible that overexpression of Lamp2A in CHO M12 cells leads to CMA activation that allows AnxA6 protein degradation. Consequently, this instigates the upregulation of Rab7-GTP levels, which triggers restoration of LE-Chol export to eventually reduce the levels of cholesterol in the LE/Lys compartment of NPC1 mutant cells.

In previous studies we demonstrated that AnxA6 is associated with acidic, endo-, and pre-lysosomal compartments of non-polarized as well as polarized cells [37,55,56]. Conversely, in settings where LE/Lys accumulate cholesterol, such as in NPC1 disease or after pharmacological NPC1 inhibition, increased amounts of AnxA6 proteins were found associated with LE/Lys [57]. The latter findings were based on colocalization studies using confocal microscopy as well as subcellular fractionation techniques, and in order to get additional insights into the AnxA6 location in the complex membrane organization of the LE/Lys compartment of NPC1 mutant cells, we compared the distribution of AnxA6 in CHO-WT and CHO M12 cells at much higher resolution, using electron microscopy. Therefore, ultrathin sections of both cell lines were labeled (for details see Methods) with an anti-AnxA6 polyclonal antibody [57] followed by anti-rabbit 12 nm colloidal gold antibodies (Figure 5A). Quantification of gold particle density (gold particles/μm^2^) or vesicle area (μm^2^) validated previous findings, identifying a significant increase of AnxA6-gold labeling in LE/Lys structures of CHO M12 compared to CHO-WT cells (Figure 5A,B). Moreover, the complex and heterogeneous inner LE/Lys structures consisting of multilamellar membranes and electrodense material in prototypical endolysosome in CHO M12 cells were enriched (~95%) with anti-AnxA6 labeling (Figure 5C, red arrowheads). Much less prominent AnxA6 labeling was also observed at the perimeter membrane of those LE/Lys structures (white arrowheads). In fact, these findings are supported by others, identifying significant levels of AnxA6 in the lysosomal matrix, compared to the corresponding lysosomal membranes isolated from liver [30].

Therefore, AnxA6 and LE-Chol seem to be engaged in a feedback loop in the LE/Lys compartment of NPC1 mutant cells (Figure 6). Upregulated AnxA6 levels increase the amount of LE-Chol, and, inversely, increased LE-Chol levels upon loss of NPC1 function raise AnxA6 protein amounts. Given that several studies pointed at LE-Chol accumulation to interfere with CMA and lysosomal function [13,45,48,58,59,60], we propose that this connectivity of LE-Chol levels and CMA efficacy is the underlying cause for the inhibition of AnxA6 degradation by selective autophagy. In support of this hypothesis, Rodriguez-Navarro et al., 2012 demonstrated that increased lipid intake in mice, either feeding a high-fat or cholesterol-rich diet, dramatically decreased Lamp2A levels due to accelerated degradation [48], causing a substantial decline in CMA activity that was associated with increased amounts of AnxA6.

Although the data presented here may favor Lamp2A-induced CMA activation being responsible for AnxA6 degradation (Figure 6), it should be emphasized that other routes, such as eMi or even specific lysosomes, may also contribute to control AnxA6 protein turnover, and cannot be entirely excluded [30,50]. Importantly, this study clearly identifies a close relationship between AnxA6 protein and LE-Chol levels in NPC1 mutant cells. Future studies, using pharmacological inhibitors to specifically block the abovementioned pathways will be needed to dissect the contribution of CMA or other degradation routes in AnxA6 protein turnover.

The underlying mechanisms that lead to an enrichment of AnxA6 protein pools on inner LE/Lys membranes and its molecular consequences remain to be clarified. AnxA6 is the largest member of the annexin family and instead of the four annexin repeats in all other annexins, has an unusual size and structure with eight annexin domains that are connected by a hinge-like loop. This provides some flexibility in the membrane interactions for AnxA6, which may be sensitive to changes in the lipid microenvironment. Within this context, alterations in the amount of LE-Chol indeed appear to induce conformational changes in proteins residing in the limiting membrane of LE/Lys, with consequences for protein-protein interactions. The LE-Chol-sensitive formation of the ORP1L-VAP-A complex, which participates in cholesterol transfer from LE to the ER, is currently the best example for this mechanism. Moreover, ORP1L-VAP-A interaction in response to alterations in LE-Chol levels impairs Rab7-dependent autophagosome maturation [63,64]. Thus, LE-Chol levels may also promote redistribution and conformational changes that impact on the ability of AnxA6, and possibly another annexin, AnxA1, to participate in the control of cholesterol transfer from LE/Lys to the ER (and vice versa) through MCS, with far reaching implications for cholesterol homeostasis [8,24,25].

Previous studies identified AnxA6 as a cholesterol-interacting protein [65,66]. This coincides with the increased association of AnxA6 with LE-Chol-enriched vesicles [57], which not only contain increased amounts of cholesterol in internal vesicles, but, as shown by several studies, also in the perimeter LE/Lys membrane [52]. Thus, the recruitment of cytosolic pools of AnxA6 proteins with an increased affinity for cholesterol in the limiting membrane of LE/Lys could in part explain these observations. Strikingly, this study identified substantial amounts of AnxA6 associated with inner membranes and ILV of the late endocytic compartment. Although it remains to be determined how AnxA6 is delivered to inner LE/Lys membranes upon LE-Chol accumulation, it is well known that cholesterol can accumulate in ILV of MVB [67]. This observation may be relevant for other biological events that commence in the LE/Lys compartment. For instance, exosomes originate from the ILV of MVB to then fuse with the plasma membrane for subsequent extracellular release. Despite the rapidly increasing knowledge regarding the complex repertoire of bioactive molecules in exosomes, including proteins, RNAs, and lipids, its biogenesis and the loading of these structures with cargo is still not well understood. Interestingly, based on the proteomic databases available on exosomes isolated from a large variety of settings, amongst all kinds of proteins, exosomes are highly enriched in annexins (ExoCarta exosome database: www.exocarta.org, accessed on 29 July 2015). Several questions arise from these observations: (1) What are the targeting mechanisms that deliver and allow access of annexins into exosomes? Could these mechanisms include an escape from CMA- or eMI-mediated degradation once annexins enter the lumen of MVBs? (2) What is the function of annexins once they are released in exosomes and enter target cells? (3) Finally, and most relevant in relation to NPC disease, do annexins and cholesterol pools deposited in exosomes have a function in cellular cholesterol homeostasis? In fact, recent studies identified drastically increased amounts of cholesterol-rich exosomes released from NPC1 mutants or U18666A-treated cells. Thus, it was proposed that upregulated exosome release of exosomes might serve to ameliorate LE-Chol accumulation [68] and consequently the NPC1 phenotype.

In summary, this study demonstrates that CMA is inhibited in NPC1 mutant cells causing an increase in AnxA6 protein levels. This inhibits Rab7 activity and its ability to overcome accumulation of cholesterol in LE/Lys, which is detrimental for cellular homeostasis. Lamp2A-mediated restoration of CMA promotes AnxA6 protein degradation and is associated with a rescue of LE-Chol accumulation in NPC1 mutant cells. Taken together, AnxA6 protein levels appear to be associated with LE-Chol accumulation and CMA-mediated protein degradation, the functionality of the latter being highly dependent on LE-Chol levels.

## Figures and Tables

**Figure 1 cells-09-01152-f001:**
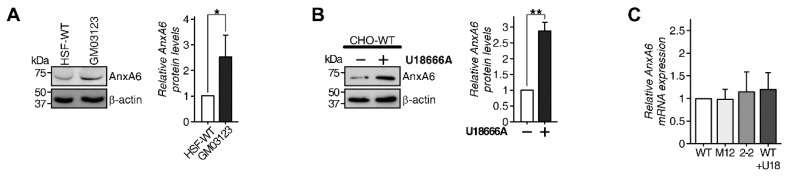
Elevated AnxA6 protein levels in NPC1 mutant cells. Cell lysates from (**A**) human control (HSF) and NPC1 patient (GM03123) skin fibroblasts and (**B**) CHO-WT treated for 16 h with and without U18666A (2 µg/mL) as indicated were analyzed by Western blotting with antibodies against AnxA6 and β-actin as loading control. Western blots representative for three independent experiments are shown (* *p* < 0.05, ** *p* < 0.01). (**C**) Relative mRNA quantification determined by qPCR of *AnxA6* in CHO-WT, CHO M12, CHO 2-2, and CHO-WT cells treated with U18666A (WT + U18) is shown. The housekeeper gene *Rpl13* served as control. Data shown represents the mean ± SEM from three independent experiments with duplicate samples.

**Figure 2 cells-09-01152-f002:**
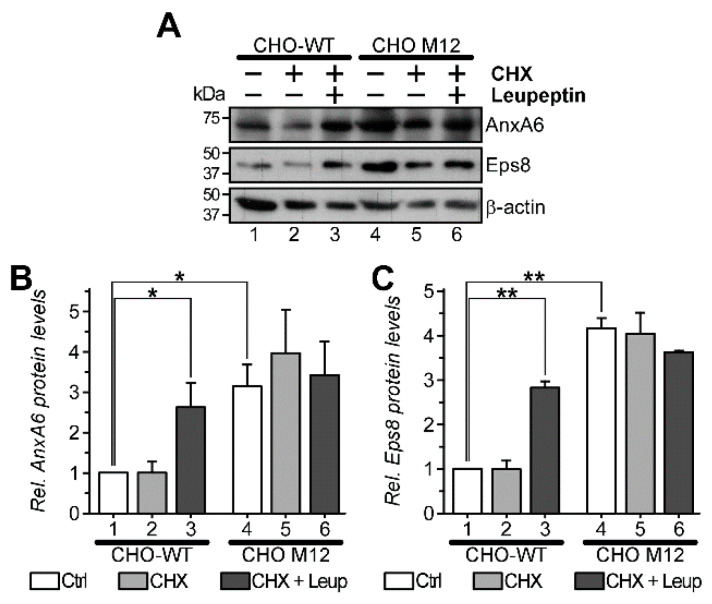
Reduced lysosomal degradation of AnxA6 and Eps8 in NPC1 mutant cells. (**A**) CHO-WT (lane 1–3) and CHO M12 (lane 4–6) cells treated without (control) or with 100 µg/mL cycloheximide (CHX) alone or together with 10 µg/mL leupeptin (CHX + Leup) as indicated for 48 h. Cell lysates were prepared and analyzed by Western blotting for AnxA6, Eps8 and β-actin levels as indicated. (**B**,**C**) The intensity of bands was densitometrically quantified. Data shown represents the mean ± SEM of two independent experiments (* *p* < 0.05, ** *p* < 0.01).

**Figure 3 cells-09-01152-f003:**
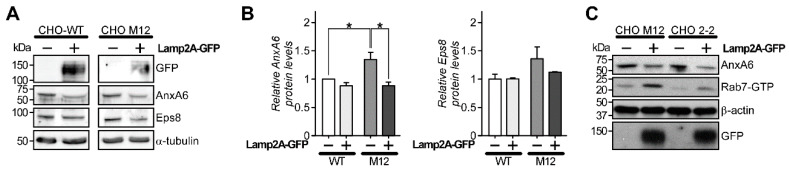
Overexpression of Lamp2A increases AnxA6 protein degradation. (**A**) Cell lysates from CHO-WT and CHO M12 cells ± ectopically expressed Lamp2A-GFP as indicated were analyzed by Western blotting for GFP, AnxA6, Eps8, and α-tubulin. A representative western blot from two independent experiments is shown. (**B**) The intensity of AnxA6 and Eps8 bands was quantified and normalized to α-tubulin. Data shown represents the mean ± SEM (* *p* < 0.05). (**C**) CHO M12 and CHO 2-2 cells were transfected ± Lamp2A-GFP as above. Cell lysates were prepared and subjected to Rab interacting lysosomal protein (RILP)-C33–GST pull-down assays to determine active Rab7 (Rab7-GTP) levels (see Material and Methods for details). Total levels of AnxA6, β-actin, and GFP in cell lysates are shown.

**Figure 4 cells-09-01152-f004:**
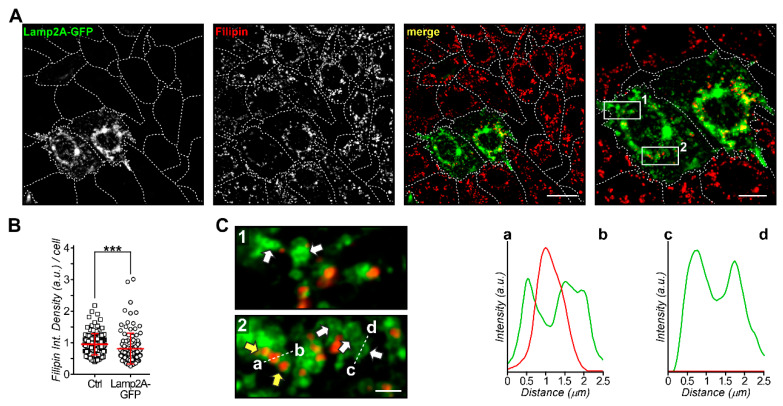
Lamp2A-GFP overexpression reduces late endosomal cholesterol (LE-Chol) accumulation in NPC1 mutant cells. (**A**) CHO M12 cells were transfected with Lamp2A-GFP (green), fixed, and stained with filipin to visualize free cholesterol (red). A representative image, including the merged image, is shown (scale bar, 20 μm; 10 μm in enlarged right panel). The shape of all cells was outlined. (**B**) The fluorescence intensity of filipin staining (integrated density per cell) of at least 20 control (Ctrl) and Lamp2A-GFP transfected cells from three independent experiments was quantified and shown in a dot plot. The mean ± SEM is also given (*** *p* < 0.001). (**C**) Two enlarged regions of interest from A (1 and 2) show details of Lamp2A-GFP/filipin colocalization and vesicle heterogeneity in transfected cells (scale bar, 2 μm). White arrows point at filipin-negative, but Lamp2A-positive vesicles (1,2) and yellow arrows highlight Lamp2A-positive vesicles colocalizing with filipin (2). Line profiles of fluorescence intensities of Lamp2A-GFP (green) and filipin (red) are shown (a–b and c–d, 2.5 μm).

**Figure 5 cells-09-01152-f005:**
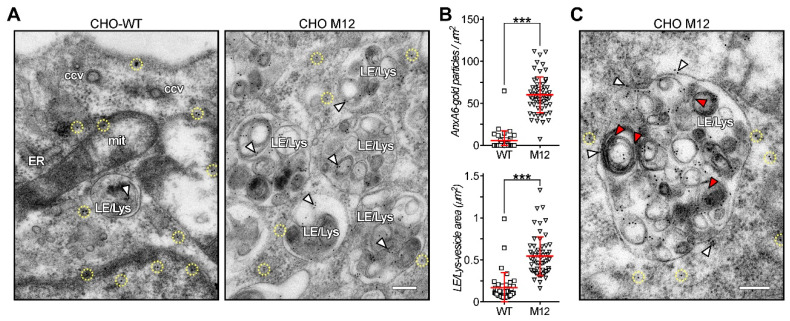
AnxA6 is enriched in inner membranes of the late endosome/lysosome (LE/Lys) compartment in NPC1 mutant cells. Lowicryl HM20 ultrathin sections of CHO-WT and CHO M12 cells were labeled with anti-AnxA6. (**A**) A region of the cytoplasm of CHO-WT and CHO M12 cells with several LE/Lys structures is shown. Note that in CHO M12 cells AnxA6 gold labeling is much more prominent compared to CHO-WT cells. Encircled gold particles (dashed yellow) indicate cytoplasmatic AnxA6 labeling. (**B**) Quantification of gold particle density (gold particle/μm^2^) and LE/Lys vesicle area (μm^2^). The data is shown as dot plot per cell. The mean ± SEM is also given (*** *p* < 0.001). (**C**) Representative enlarged image of a prototypical endolysosome in CHO M12 cells. AnxA6 labeling is enriched in inner LE/Lys structures (red arrowheads) and less prominent at the LE/Lys perimeter (white arrowheads). mit, mitochondria; ccv, clathrin coated vesicles; ER, endoplasmic reticulum. Scale bar is 200 nm.

**Figure 6 cells-09-01152-f006:**
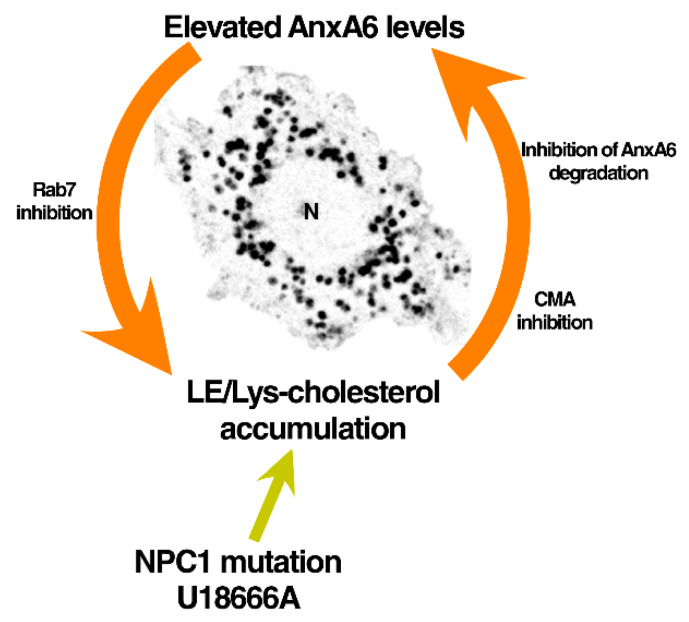
Scheme depicting the feedback loop of AnxA6 and LE-Chol levels in NPC1 mutant cells. NPC1 loss-of-function mutations or pharmacological NPC1 inhibition (U18666A) lead to late endosome/lysosome (LE/Lys)-cholesterol accumulation [1,6,61]. This interferes with Lamp2A-dependent chaperone-mediated autophagy (CMA) [31,48], which leads to reduced AnxA6 protein degradation (this study). Elevated AnxA6 protein levels promote Rab7 inactivation, which further contributes to cholesterol accumulation in LE/Lys (this study and [25]). In fact, diminution of active Rab7 in NPC1 mutant cells could also inhibit CMA [62]. This inhibitory feedback loop can be overcome by Lamp2A overexpression, which activates CMA [49], leading to reduced AnxA6 levels and concomitantly, also induces the diminution of LE/Lys-cholesterol accumulation (this study). Lamp2A-induced restoration of LE-cholesterol export being responsible for reduced AnxA6 protein levels cannot be completely ruled out. Alternatively, AnxA6 depletion in NPC1 mutant cells increased Rab7 activity [25], which restored cholesterol export from LE/Lys. Taken together, this feedback loop connects cholesterol levels in LE/Lys with CMA-mediated AnxA6 protein degradation, with CMA functionality being highly dependent on cholesterol levels in the LE/Lys compartment. N, nucleus.

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
