# Peer review of "Selective Degradation Permits a Feedback Loop Controlling Annexin A6 and Cholesterol Levels in Endolysosomes of NPC1 Mutant Cells"

_cells, 2020, doi:10.3390/cells9051152_

Round 1

Reviewer 1 Report

The present study investigated the molecular mechanism for enhanced AnxA6 protein levels in NPC1 mutant cells. The authors found that AnxA6 protein is degraded through the CMA pathway, and showed that Lamp2A-mediated restoration of CMA in NPC1 cells reduces LE-Chol accumulation in NPC1 mutant cells. This study showed a role of AnxA6 for LE-Chol accumulation in NPC1 disease and might be of interest for researchers in the field of Annexin biology. However, the paper needs a considerable revision for publication, and some experiments need to be repeated to confirm the reproducibility of the results.

Major concerns

  • Figure 4 is missing from the paper. According to the results description, the data shown in Figure S1 are supposed to be Figure 4, and the data of Figure S1 are not shown.
  • There are no error bars on the graphs in Figure 2B, Figure 2C and the left panel of Figure 3C.
  • In Figure 2, the protein levels of AnxA6 and Eps8 are greater than those of Ctrl even in the presence of CHX in CHP-WT cells. I cannot understand the AnxA6 and Eps8 proteins accumulated in the absence of de novo protein synthesis. This needs to be addressed.
  • The experiment shown in Figure 3 must be repeated and the results should be confirmed. Although the intensity of the AnxA6 band in the absence of Lamp2A was significantly greater in CHO M12 cells than that in CHO-WT cells, the protein levels in CHO-WT cells seem to be similar to those in CHO M12 cells in Figure 3A. Similarly, although the intensity of the AnxA6 was not decreased by the Lamp2A-GFP overexpression in CHO-WT cells, the protein level of AnxA6 clearly decreased by the Lamp2A-GFP expression in CHO-WT cells in Figure 3A.
  • The results described that the Rab7-GTP levels were elevated in CHO M12 and CHO 2-2 cells (lines 319-320), the data of control (CHO-WT cells) were not shown in Figure 3C.
  • A photomicrograph of CHO-WT cells is needed in Figure 5A.

Minor comments

  • The statements in lines 240-247 and 269-275 are confusing and difficult to follow.
  • The legend for Figure 6 should be shortened.
  • There is a gap between line 345 and 346.

Author Response

Reviewer #1 (Comments and Suggestions for Authors):

The present study investigated the molecular mechanism for enhanced AnxA6 protein levels in NPC1 mutant cells. The authors found that AnxA6 protein is degraded through the CMA pathway, and showed that Lamp2A-mediated restoration of CMA in NPC1 cells reduces LE-Chol accumulation in NPC1 mutant cells. This study showed a role of AnxA6 for LE-Chol accumulation in NPC1 disease and might be of interest for researchers in the field of Annexin biology. However, the paper needs a considerable revision for publication, and some experiments need to be repeated to confirm the reproducibility of the results.

Major concerns:

  1. Figure 4 is missing from the paper. According to the results description, the data shown in Figure S1 are supposed to be Figure 4, and the data of Figure S1 are not shown.

As rightly pointed out by this reviewer, we apologize for the confusion created by the wrong positioning and mislabelling of figures. Unfortunately, during the editing of our originally submitted manuscript by the MDPI production department, the Figure 4 was mislabelled as Figure S1 and the original Figure S1 (incl. Figure legend S1) was completely missing from the version send out for review. As outlined below, this also created some confusion for the other Reviewers (Reviewer #3 and #4) and is clarified in more detail below.

Figure 4 was mislabelled and shown as Figure S1 on page 7 already, but clearly and alike our originally submitted manuscript, should have been placed after Figure 3. In addition, the original Figure S1 and the matching Figure Legend S1, which should both have been at the end of the manuscript, were completely missing. Please see the proper arrangement and labelling of these figures in the new revised version attached with this letter.

  1. There are no error bars on the graphs in Figure 2B, Figure 2C and the left panel of Figure 3C.

We apologize for the lack of information in these figures. As requested by the reviewer, we searched in old notebooks of PhD students that performed these experiments and now provide statistical analysis and the mean ± SEM in Figure 2B-C and both panels of Figure 3B. In fact, in Figure 2B-C, AnxA6 and Eps8 levels in CHO-WT cells treated with CHX + leupeptin (compare lane 1 and 3) and CHO M12 cells (compare lane 1 and 4) were significantly elevated (* p <0.05; ** p <0.01), respectively. Likewise, in Figure 3B, AnxA6 levels were significantly elevated in CHO M12 compared to CHO-WT cells, confirming data sets from Figure 2 (* p <0.05). Yet, ectopic expression of Lamp2-GFP significantly reduced AnxA6 levels in CHO M12 cells (* p <0.05; see left panel in Figure 3B). In addition, the number of replicates is provided for all experiments in both figure legends. Importantly, the inclusion of these data sets that were added to the quantifications did not change the overall outcome and interpretation of these studies.

  1. In Figure 2, the protein levels of AnxA6 and Eps8 are greater than those of Ctrl even in the presence of CHX in CHP-WT cells. I cannot understand the AnxA6 and Eps8 proteins accumulated in the absence of de novo protein synthesis. This needs to be addressed.

AnxA6 and Eps8 levels in CHO-WT cells incubated with either CHX or CHX + leupeptin were quantified in 2 independent experiments. AnxA6 and Eps8 levels in CHO-WT cells incubated with CHX alone were not elevated compared to the control, which is evident from visual inspection of the western blot (compare lane 1 and 2). Moreover, as shown in the densitometric quantification of western blots in Fig. 2B and 2C, only when CHO-WT cells were incubated with both inhibitors, an increase in AnxA6 and Eps8 protein levels was observed (compare lane 1 and 3; * p <0.05; ** p <0.01), indicating inhibition of lysosomal degradation. This significant increase of AnxA6 and Eps8 protein levels in the presence of leupeptin was not observed in CHO M12 cells (lane 4-6 in Fig. 2A, for quantification see Fig. 2B-C), implicating compromised lysosomal degradation in this NPC1 mutant cell line.

  1. The experiment shown in Figure 3 must be repeated and the results should be confirmed. Although the intensity of the AnxA6 band in the absence of Lamp2A was significantly greater in CHO M12 cells than that in CHO-WT cells, the protein levels in CHO-WT cells seem to be similar to those in CHO M12 cells in Figure 3A. Similarly, although the intensity of the AnxA6 was not decreased by the Lamp2A-GFP overexpression in CHO-WT cells, the protein level of AnxA6 clearly decreased by the Lamp2A-GFP expression in CHO-WT cells in Figure 3A.

As requested by the reviewer, we now provide the quantification (mean ± SEM) of relative AnxA6 and Eps8 protein levels from two independent experiments in the two panels of Figure 3B from the western blot analysis shown in Figure 3A. Quantification of these data sets in Figure 3B identified significantly elevated AnxA6 levels in CHO M12 compared to CHO-WT cells, confirming data sets from Figure 2 (* p <0.05) and is consistent with our recent study (Ref #25). Yet, ectopic expression of Lamp2-GFP significantly reduced AnxA6 levels in CHO M12 cells (* p <0.05; see left panel in Figure 3B). We agree with this reviewer that elevated AnxA6 levels in CHO M12 cells cannot readily be identified by visual inspection of the western blots shown in Figure 3A and might not be as evident is in Figure 2A-B, or our recent publication in Cell Mol Life Sci (Ref. #25). However, it is important to note that the quantification of AnxA6 protein levels was normalized to a-tubulin levels for each experiment, and with this correction the comparison of AnxA6 levels in CHO-WT and CHO M12 cells was indeed statistically significant.

  1. The results described that the Rab7-GTP levels were elevated in CHO M12 and CHO 2-2 cells (lines 319-320), the data of control (CHO-WT cells) were not shown in Figure 3C.

Rab7 activity in CHO cells with moderate (CHO-WT) or elevated (e.g. CHO M12, CHO 2-2) AnxA6 protein levels, as judged by the determination of Rab7-GTP levels in pulldown assays, has been examined by our laboratories in great detail recently (Ref. #25). In these studies, CHO M12 and 2-2 cells with elevated AnxA6 protein expression displayed a substantial and significant reduction in Rab7-GTP levels compared to CHO-WT cells. Likewise, stable or transient AnxA6 overexpression in CHO-WT cells was accompanied by reduced Rab7-GTP levels (Ref. #25). On the other hand, AnxA6 knockdown in CHO M12 or CHO 2-2 cells was associated with effectively increased Rab7-GTP levels. This correlated with reduced cholesterol accumulation in AnxA6-depleted CHO M12 and 2-2 cells (Ref. #25).

Lamp2-GFP overexpression has been described to rescue LE-cholesterol accumulation upon NPC1 inhibition (Refs. #52-54), which we also confirmed in this study (Figure 4). This provided the unique opportunity to examine if the Lamp2-induced rescue of the NPC1 phenotype was associated with a reduction in AnxA6 protein levels and concomitant elevation of Rab7 activity. In contrast, CHO-WT cells do not display late endosomal (LE) cholesterol accumulation under standard growth conditions (10% FCS) and were therefore not suitable to study rescue of LE-cholesterol accumulation in NPC1 mutant models upon Lamp2A-GFP overexpression by microscopy in Figure 4. We therefore reasoned that CHO-WT cells were not appropriate and relevant in this context to assess the impact of Lamp2A overexpression on Rab7 activity and provide valuable information that could link to late endosomal cholesterol homeostasis and lysosomal degradation under dysregulated (NPC1 mutation) conditions.

  1. A photomicrograph of CHO-WT cells is needed in Figure 5A.

We apologize for this oversight and have corrected Figure 5 accordingly. The panel A in Figure 5 now shows photomicrographs of both CHO-WT and CHO M12 cells. The panel B now shows the quantification of gold-particle density (gold particle/μm2) and LE/lysosome vesicle area (μm2) and a representative image of a prototypical endolysosome from CHO M12 cells. The figure legend was changed accordingly. 

Minor concerns:

  1. The statements in lines 240-247 and 269-275 are confusing and difficult to follow.

We apologise for the difficulty to follow these sections, which refers to data described in Figure S1, which was completely missing in the version send out for review due to the errors that occurred during the editing of our originally submitted manuscript by the MDPI production department (see comments for major point 1 above). Hence, it would have been difficult to understand and follow the flow of the manuscript, as the relevant data was missing.

Furthermore, for a better understanding, we revised and shortened the section in lines 269-275 (now lines 269 – 275).

  1. The legend for Figure 6 should be shortened.

As requested by this reviewer, to avoid repetitiveness, the figure legend of Figure 6 was shortened in the revised version of the manuscript.

  1. There is a gap between line 345 and 346.

We apologize for this error, yet we were unable to notice any gap between these two lines in the version that was available to us after download from the MDPI website. However, in the version that we received from MDPI a gap appeared between lines 352 and 353. As there was no text missing, we assume this to be an issue specific to the version this reviewer received.

Reviewer 2 Report

This is a well written paper.

The major concern of the manuscript is that no experiments were done to measure the levels of cholesterol inside the cells. Without direct measuring the levels of cholesterol under the proposed different experimental conditions, the conclusion would be premature and speculative.

Author Response

Reviewer #2

Comments and Suggestions for Authors

This is a well written paper.

The major concern of the manuscript is that no experiments were done to measure the levels of cholesterol inside the cells. Without direct measuring the levels of cholesterol under the proposed different experimental conditions, the conclusion would be premature and speculative.

We apologize for the lack of explanation regarding the quantification of late endosomal (LE) cholesterol in CHO M12 cells ± Lamp2A-GFP, which is shown in Figure 4B and represents the quantification of filipin staining of whole cells (³ 20 cells per group) shown in Figure 4A. Filipin binds free cholesterol and is commonly used to quantify the relative amounts of cholesterol in cells. As outlined in more detail below, this approach is widely used and most relevant when aiming to analyze changes in the relative amounts of LE-cholesterol in models related to NPC1 disease.

As shown in Figure 4A, LE-cholesterol accumulation in NPC1 mutant cells is generally identified by massive filipin staining in enlarged, perinuclear vesicles. In these cells, filipin staining of the plasma membrane, which in normal cells represents up to 90% of cellular cholesterol, is usually very weak and redundant, making it impractical to utilize this methodology for the visualization and quantification of cholesterol at the plasma membrane or other cellular organelles in NPC1 mutant models. Hence, because of this major cholesterol redistribution in NPC1 disease, quantification of filipin staining within whole NPC1 mutant cells is considered a highly appropriate approach to determine relative amounts of LE cholesterol (Höltta-Vuori et al., Mol Cell Biol 2002).

Yet, as shown by our laboratories (Ref. #27), but also other world-leading researchers in this field (Refs #1, #4), when based on biochemical approaches, this reflects a rather moderate increase of LE-cholesterol or total cellular cholesterol (Sokol et al., J Biol Chem 1988; Frolov et al., J Biol Chem 2003; Puri et al., J Biol Chem 2003; Narita et al., Faseb J 2005). This is further highlighted by the fact that only 10-30% of total cellular cholesterol is found in endosomes (Ikonen E Physiol Rev 2006). For instance, when studying a cell model for lysosomal storage diseases, Puri et al. (Puri et al., J Biol Chem 2003) reported a 25-35 % increase in cholesterol levels biochemically, corresponding to massive increases in filipin staining.

Several groups, including our laboratories, have studied approaches to rescue cholesterol accumulation in NPC1 mutant cells. These studies were often based on the transient overexpression (e.g. Rab9, Rab7, Lamp2A) (e.g. Höltta-Vuori et al., Curr Biol 2000 and Ref. #54) or knockdown of genes (AnxA6) (Ref#25). In these complex experimental settings, and given the limitations to determine LE-cholesterol biochemically, filipin staining, ideally in combination with a LE-marker (e.g. Rab7, Lamp2A), proved to be the most reliable methodology to document changes in cholesterol levels in the LE/Lys compartment. For example, Narita et al. showed a strong reduction of filipin staining in NPC mutant cells after overexpression of Rab9 (Narita et al., Faseb J 2005). These findings only corresponded to a 20 % reduction of free cholesterol measured biochemically. Likewise, based on the quantification of filipin staining, Lamp2A overexpression reduced LE-cholesterol accumulation upon incubation with different doses of the pharmacological NPC1 inhibitor U18666A by 10-40% (Ref. #54). Similarly, in Lamp2-deficient mouse embryo fibroblasts, filipin staining was the method of choice to demonstrate that restoration of Lamp2 expression in these cells reduced LE-cholesterol accumulation (Ref. #53). Also, in our recent study, we identified a ~35 % reduction in filipin staining of LE vesicles in AnxA6-depleted NPC1 mutant cells (Ref. #25).

To clarify this point, the following was added in the revised manuscript: Quantification of filipin staining in whole NPC1 mutant M12 cells, which resembles almost exclusively prominent and enlarged, perinuclear filipin- and Lamp2A-positive vesicles see also enlarged inserts and quantification in Figure 4B) identified a reduction in the accumulation of LE-cholesterol by 15 % in CHO M12 cells (9.474 ±0.3 vs. 8.059 ±0.45) ectopically expressing Lamp2A cells compared to neighboring control cells.

Reviewer 3 Report

This paper discusses a new relationship between Annexin A6 and NPC proteins. The idea of this study is very impressive and the results are also very eye-catching. However, I got the impression that the sentences were generally redundant. Also, I think it is necessary to review the whole sentence such as mistakes in Figure numbers.

Author Response

Reviewer #3

Comments and Suggestions for Authors

This paper discusses a new relationship between Annexin A6 and NPC proteins. The idea of this study is very impressive and the results are also very eye-catching. However, I got the impression that the sentences were generally redundant. Also, I think it is necessary to review the whole sentence such as mistakes in Figure numbers.

We apologize for errors that occurred during the MDPI editing of our original submission, which is explained in more detail above (Reviewer #1, point 1; Reviewer #3, point 2-3). The whole manuscript has been revised accordingly, and in the new version, all figures are present and appear in the right order. Moreover, to provide a better understanding, we have revised the text body and shortened several sections as outlined above (Reviewer #1, minor points 1-2).

[Reviewer #1, point 1: As rightly pointed out by this reviewer, we apologize for the confusion created by the wrong positioning and mislabelling of figures. Unfortunately, during the editing of our originally submitted manuscript by the MDPI production department, the Figure 4 was mislabelled as Figure S1 and the original Figure S1 (incl. Figure legend S1) was completely missing from the version send out for review. As outlined below, this also created some confusion for the other Reviewers (Reviewer #3 and #4) and is clarified in more detail below.

Figure 4 was mislabelled and shown as Figure S1 on page 7 already, but clearly and alike our originally submitted manuscript, should have been placed after Figure 3. In addition, the original Figure S1 and the matching Figure Legend S1, which should both have been at the end of the manuscript, were completely missing. Please see the proper arrangement and labelling of these figures in the new revised version attached with this letter.]

Reviewer 4 Report

The study performed by Dr. Enrich’s group has investigated the association between AnxA6 protein levels and the extent of LE-Cholesterol accumulation in the LE/Lys compartment of the NPC1 mutant cells. Their findings reveal a complex feedback loop involving CMA that controls AnxA6 and LE-Cholesterol levels. The main conclusion is supported by data and the manuscript is technically sound.

In figure 2, the authors show lower lysosomal degradation of AnxA6 in NPC1 mutant cells, but it is not clear how many replicates they use for these experiments. Is this a representative experiment? I am rather concerned about the variability of this experiment. This point should be addressed.

Figure S1 is a new figure (2) or a supplementary figure? In this case, it should be included in supplementary material.

Figure 4 should be included in the main manuscript’s text. This image appears to have important information on cholesterol distribution.

Author Response

Reviewer #4

Comments and Suggestions for Authors

The study performed by Dr. Enrich’s group has investigated the association between AnxA6 protein levels and the extent of LE-Cholesterol accumulation in the LE/Lys compartment of the NPC1 mutant cells. Their findings reveal a complex feedback loop involving CMA that controls AnxA6 and LE-Cholesterol levels. The main conclusion is supported by data and the manuscript is technically sound.

  1. In figure 2, the authors show lower lysosomal degradation of AnxA6 in NPC1 mutant cells, but it is not clear how many replicates they use for these experiments. Is this a representative experiment? I am rather concerned about the variability of this experiment. This point should be addressed.

We apologize for the lack of information in this figure. As requested by this reviewer (and Reviewer #1), we searched in old notebooks of PhD students that performed these experiments and now provide statistical analysis and the mean ± SEM in Figure 2B-C. As pointed out in detail above (see response to Reviewer #1, point 3), the quantification of data shown in Figure 2A represents the mean ± SEM from two independent experiments. The number of replicates is now also provided in the figure legend. AnxA6 and Eps8 levels in CHO-WT cells incubated with leupeptin showed a significant increase in AnxA6 and Eps8 protein levels (compare lane 1 and 3; * p <0.05; ** p <0.01), indicating inhibition of lysosomal degradation. This significant increase of AnxA6 and Eps8 protein levels in the presence of leupeptin was not observed in CHO M12 cells (lane 4-6 in Figure 2A, for quantification see Figure 2B-C), implicating compromised lysosomal degradation in this NPC1 mutant cell line. Hence, the inclusion of the data sets that were added to the quantifications did not change the overall outcome and interpretation of this experiment.

  1. Figure S1 is a new figure (2) or a supplementary figure? In this case, it should be included in supplementary material.

 As rightly pointed out by this reviewer (and Reviewer #1, see point 1), the misplaced and wrongly labelled figure created confusion amongst Reviewers #1, #3 and #4. Unfortunately, during the editing of our originally submitted manuscript by the MDPI production department, the Figure 4 was mislabelled and shown as Figure S1 on page 7 already, but clearly and alike our originally submitted manuscript, should have been placed after Figure 3. In addition, the original Figure S1 and the matching Figure Legend S1, which should both have been at the end of the manuscript, were completely missing from the version that was send to the reviewers. Please see the proper arrangement and labelling of these figures in the new revised version attached with this letter.

  1. Figure 4 should be included in the main manuscript’s text. This image appears to have important information on cholesterol distribution.

See response to point 2 of this Reviewer above (and Reviewer #1, point 1).

[Reviewer #1, point 1: As rightly pointed out by this reviewer, we apologize for the confusion created by the wrong positioning and mislabelling of figures. Unfortunately, during the editing of our originally submitted manuscript by the MDPI production department, the Figure 4 was mislabelled as Figure S1 and the original Figure S1 (incl. Figure legend S1) was completely missing from the version send out for review. As outlined below, this also created some confusion for the other Reviewers (Reviewer #3 and #4) and is clarified in more detail below.

Figure 4 was mislabelled and shown as Figure S1 on page 7 already, but clearly and alike our originally submitted manuscript, should have been placed after Figure 3. In addition, the original Figure S1 and the matching Figure Legend S1, which should both have been at the end of the manuscript, were completely missing. Please see the proper arrangement and labelling of these figures in the new revised version attached with this letter.]

Round 2

Reviewer 1 Report

The authors have addressed all of the comments raised by this reviewer. I would like to recommend that the manuscript is acceptable for publication.

Reviewer 2 Report

The authors have addressed the concerns. The results presented would be of interest to the scientific community.